# A Review of the Current Clinical Evidence for Loco-Regional Moderate Hyperthermia in the Adjunct Management of Cancers

**DOI:** 10.3390/cancers15020346

**Published:** 2023-01-05

**Authors:** Brendan Seng Hup Chia, Shaun Zhirui Ho, Hong Qi Tan, Melvin Lee Kiang Chua, Jeffrey Kit Loong Tuan

**Affiliations:** 1Division of Radiation Oncology, National Cancer Centre Singapore, 11 Hospital Drive, Singapore 169610, Singapore; 2Department of Radiation Oncology, 585 North Bridge Rd, Level 10 Raffles Specialist Centre, Singapore 188770, Singapore

**Keywords:** hyperthermia, locoregional moderate hyperthermia, electro hyperthermia, cancer management, cancer treatment, review, complementary therapy

## Abstract

**Simple Summary:**

There is a large gap in knowledge amongst the oncology community of moderate hyperthermia use in cancer management. This review provides an overview of clinical data on the use of loco-regional and superficial hyperthermia in the adjunct management of cancers. It is updated using higher-level evidence from prospective, comparative studies and meta-analyses. The methodology and results are summarised and tabulated according to tumour type for easy reference.

**Abstract:**

Regional hyperthermia therapy (RHT) is a treatment that applies moderate heat to tumours in an attempt to potentiate the effects of oncological treatments and improve responses. Although it has been used for many years, the mechanisms of action are not fully understood. Heterogenous practices, poor quality assurance, conflicting clinical evidence and lack of familiarity have hindered its use. Despite this, several centres recognise its potential and have adopted it in their standard treatment protocols. In recent times, significant technical improvements have been made and there is an increasing pool of evidence that could revolutionise its use. Our narrative review aims to summarise the recently published prospective trial evidence and present the clinical effects of RHT when added to standard cancer treatments. In total, 31 studies with higher-quality evidence across various subsites are discussed herein. Although not all of these studies are level 1 evidence, benefits of moderate RHT in improving local tumour control, survival outcomes and quality of life scores were observed across the different cancer subsites with minimal increase in toxicities. This paper may serve as a reference when considering this technique for specific indications.

## 1. Introduction

Therapeutic hyperthermia (HT) encompasses the application of heat to targeted locations to increase the therapeutic response of oncological treatments. Various heating methods include direct (e.g., intracavitary and whole-body waterbed), infrared, perfusional (e.g., isolated limb perfusion, intravesical and intraperitoneal), nanoparticles, ultrasound and regional radiofrequency (RF) radiation [1]. Moderate HT is usually described at a range of 39–44 °C and its biological effects have been summarised previously and described in Figure 1 [2,3,4].

With the proposed mechanisms, synergisms with conventional treatments, such as radiotherapy (RT), chemotherapy (CT) and immunotherapy, should exist. Unfortunately, although positive results have been reported [5,6], robust clinical data remain elusive and marred by early negative trials [7,8,9]. Avid HT practitioners argue that the reasons for the hindered progress in this field are not the lack of efficacy, but the lack of funding, limited access, poorer tolerance of older technology, lack of quality assurance processes, poor temperature monitoring and heterogenous practices [1,6,10,11].To advance the field, international groups such as the European Society of Hyperthermic Oncology (ESHO) and Society of Thermal Medicine (STM) have been formed, with the aim of promoting scientific knowledge and facilitating cooperative research. Quality assurance guidance [12,13,14,15] has also been published this past decade, to provide technical standardisations for the clinical applications of RHT. These serve to ensure the appropriate use of Regional Hyperthermia (RHT) and establish treatment standards to improve clinical outcomes.

RHT technology uses a capacitive or radiative system [16], whereby antennas are externally applied over a target region. Non-ionising electromagnetic radiowaves or microwaves, using different frequencies and energy, are directed towards the tumour, where energy is deposited and converted into heat. Heat distribution is calculated and the target temperatures are monitored in real-time by minimally invasive thermometric probes. An adequate temperature rise is important to achieve a good clinical outcome and a dose–effect relationship has been reported in many studies [17,18,19]. Thermal dosimetry is, thus, quantified by temperature and duration and expressed in cumulative equivalent minutes at a temperature of 43 °C (CEM43) [20] and T_X_, which represents the temperature exceeded by X% of the intra-tumour points. A variation in RHT, known as modulated electro hyperthermia (mEHT), uses non-homogeneous heating and supposedly selectively targets cancer cells. This is claimed to cause additional destabilisation of malignant cell membranes and is quite popular in use [21].

In recent decades, engineering enhancements, treatment protocols and a better understanding of thermal physiology have advanced RHT. Unfortunately, it remains obscure in the general oncology community. In this article, we review and summarise the available higher-level clinical evidence that has been published since 2000 to provide an updated clinical reference on the use of RHT with contemporary cancer treatments.

## 2. Method

A literature review was performed using PUBMED on articles that included externally applied, focused, and moderate RHT. Only full-text English articles from prospective, comparative studies, meta-analyses and systematic reviews with a publication date from January 2000 to November 2022 were used. Referenced and linked articles were included if relevant. Trials included in the meta-analysis were not re-presented to avoid duplication.

## 3. Results

### 3.1. Cervical Cancer 

A Cochrane Systematic Review that compared RT alone vs. HT + RT was performed by Lutgens [22]. A total of 6 randomised controlled trials (RCTs) [7,23,24,25,26,27,28] that comprised 487 patients with locally advanced cervical cancers (LACC) were analysed. In total, 74% of patients were FIGO stage IIIB. The complete response (CR) rate (relative risk (RR) 0.56; *p* < 0.001), local recurrence rate (hazards ratio (HR) 0.48; *p* < 0.001) and overall survival (OS) (HR 0.67; *p* = 0.05) were significantly better with combined HT + RT. No significant difference was observed in the treatment-related acute (RR 0.99; *p* = 0.99) or late grade 3–4 toxicity (RR 1.01; *p* = 0.96). In 2016, another meta-analysis [29] using updated trial data [7,23,24,25,26,27] showed continued improvements in CR (+22.1%) and loco-regional control (LRC) (+23.1%) with HT; however, the survival advantage (+8%) was no longer significant. 

A 2019 network meta-analysis (NMA) by Datta compared the effectiveness and safety of 13 various interventional techniques for LACC [30]. 9894 patients were analysed across 59 trials, including 1 trial that compared HT + CTRT vs. CTRT [31], 1 trial that compared HT + RT vs. CTRT [32] and 4 trials that compared HT + RT vs. RT [23,24,25,26,33]. A corresponding surface under the cumulative ranking curve (SUCRA) analysis was performed to objectively rank the treatment options. The top three interventions for long-term LRC were as follows: HT + RT, CTRT + adjuvant CT and HT + CTRT. The top three interventions for OS were as follows: CTRT (3-weekly cisplatin), HT + CTRT and CTRT (not cisplatin). The three best treatment options for all endpoints (OS, LRC, grade ≥3 acute and late morbidity) were HT + RT, HT + CTRT and CTRT (3-weekly cisplatin). 

More recently, Yea conducted a meta-analysis comparing radical HT + CTRT vs. CTRT alone [34]. In addition, 2 RCTs [31,35] included 536 patients with LACC. Both trials used a RF capacitive heating device (Thermotron RF-8 and NRL-004 device). Harima reported a better CR with HT in his trial (odds ratio (OR) 3.993; *p* = 0.047), although OS, disease-free survival (DFS) and local relapse-free survival (LRFS) improvements were not significant [31]. Wang reported a 5-year OS improvement (81.9% vs. 72.3%, *p* = 0.04), although LRFS was not significantly improved [35]. In the combined trial data, 5-year OS (HR 0.67; *p* = 0.03) was better in the group that received HT, although the LRFS improvement remained not statistically significant (HR 0.74; *p* = 0.16) [34]. The toxicity rates were not different between the arms. Amongst the patients who received HT, a higher CEM43T90 (≥1 min) was associated with better LRFS [17]. 

An ongoing phase III RCT that compared mEHT + CTRT vs. CTRT for 210 LACC patients was reported [36]. mEHT was given by an EHY2000 Oncothermia device. At 6 months, the odds ratios (OR) for achieving local disease control (LDC) and LRFS were 0.39 (*p* = 0.006) and 0.36 (*p* = 0.002), respectively, favouring mEHT + CTRT [36]. In addition, 2- and 3-year disease-free survival (DFS) was significantly improved by mEHT (HR 0.67; *p* = 0.017 and HR 0.70; *p* = 0.035, respectively). However, 3-year OS was not significantly improved (HR 0.72; *p* = 0.74), except for those with stage III disease (HR 0.62; *p* = 0.040) [37]. Furthermore, 16.2% of participants who received mEHT reported early grade 1–2 adverse events (AEs) (adipose tissue burns, surface burns and pain), which were resolved after 3 months. There were no grade ≥3 AEs reported. Late AEs between the arms were similar. At 6 weeks, the mEHT group reported better quality of life (QoL) outcomes and better 3-month pain and fatigue scores [38]. QoL (specifically cognitive function and pain) at 2 years was significantly improved in the mEHT group. Cost-effective analysis reported mEHT+ CTRT as superior to CTRT alone, reducing the high cost of recurrent or progressive disease (PD) [37]. Interestingly, in 108 participants who underwent 18F-FDG PET/CT scans before and at 6 months post-treatment, a significantly more complete metabolic resolution (CMR) was observed in the initial PET avid lymph nodes (LN) outside the RT field (24.1% vs. 5.6%; *p* = 0.013), suggesting a potentiation of the abscopal effect with mEHT [39].

For recurrent cervical cancers in the pelvis following previous irradiation, Lee [40] compared CT vs. CT + mEHT (EHY2000) alone in a non-randomised cohort of 38 patients. The overall response rate (ORR) improved with mEHT (72.2% vs. 40%; *p* = 0.0461). No difference in OS or toxicity was noted.

The cervical cancer articles reviewed above are summarized in Table 1. We also highlight a recent review by Ijff et al. that provides further explanation and guidance on the use of RHT in LACC [41].

### 3.2. Breast Cancer

Datta [41] performed a meta-analysis of eight trials, comparing RT vs. HT + RT (five were RCTs [8,42]) in 627 locoregional recurrent breast cancer patients. Improvement in CR was noted with HT (60.2% vs. 38.1%, RR 1.57; *p* < 0.0001). Survival data were not reported. The mean acute and late grade ¾ toxicity with RT + HT was 14.4% and 5.2%, respectively.

Loboda reported on 200 stage IIB–IIIA breast cancer patients randomized to neoadjuvant (NA) CT vs. NACT + HT [43]. Electromagnetic HT was given using the inductive MagTherm device. The patients that had HT experienced a greater average reduction in primary tumour size (31.24% vs. 22.95%; *p* = 0.034), while the ORR increased by 15.9% (*p* = 0.034) and axillary LN regression improved by 14.17% (*p* = 0.011). The post-treatment viable tumour volume was lower if patients received HT and the proportion of women eligible for breast-conserving and reconstructive surgery increased by 13.63%. The 10-year OS was higher (*p* = 0.009) in patients who underwent NACT + HT.

### 3.3. Lung Cancer

A multi-institutional IAEA conducted RCTs in 80 LA non-small-cell lung cancer (NSCLC) patients, comparing RT + HT vs. RT alone [44]. HT was given using the capacitive RF-8 Thermotron device. There were no significant differences between the arms for the local response rate or OS. However, local progression-free survival (PFS) was significantly better with HT (*p* = 0.036; 1-year PFS 29.0% vs. 67.5%) and toxicity was generally mild, with no grade 3 late toxicities.

Two RCTs reported outcomes in patients with refractory advanced NSCLC. Shen [45] randomised 80 patients to HT + CT vs. CT alone. An HY7000 RF HT device was used. No difference in the response rates was observed. However, QoL improvements were significantly better in the HT + CT group (82.5% vs. 47.5%; *p* < 0.05), especially for pain improvement. Ou [46] explored the efficacy of intravenous vitamin C with mEHT against best supportive care (BSC) in 97 patients. The 3-month disease control rate was better in the experimental arm (42.9% vs. 16.7%; *p* < 0.05). A prolonged median PFS (3 vs. 1.85 months; *p* < 0.05) and OS (9.4 vs. 5.6 months; *p* < 0.05) were noted and improved QoL scores were also observed with mEHT. The exploration of inflammatory markers showed differences in IL-6 and CRP levels after mEHT, although TNFa remained unchanged, suggesting some immune effect.

Regarding small-cell lung cancer (SCLC), Lee [47] reported the results of a prospective case–control study with 31 patients (23 CT + mEHT; 8 CT alone). mEHT was given by an EHY2000 device. A significantly enhanced survival rate was noted with mEHT (*p* < 0.02).

The breast and lung cancer articles reviewed above are summarized in Table 2.

### 3.4. Oesophageal Cancers

Hu et al. [48] performed a meta-analysis of 19 RCTs (three RCTs [49,50,51] had full texts available), comprising 1519 patients with locally advanced oesophageal cancers. Patients were randomly assigned into HT + CTRT, CTRT and/or RT groups. Comparison between HT + CTRT and CTRT showed improved 1-, 3-, 5- and 7-year survival rates (OR 1.79, 1.91, 9.99 and 9.49, respectively; *p* < 0.05) with HT. No differences in the recurrence or distant metastasis rate were noted. HT + CTRT was significantly superior in terms of CR (OR 2.00; *p* < 0.00001) and total effective rates (TER) (OR 3·47; *p* < 0.00001). Surprisingly, the observed gastrointestinal toxicities were less with HT + CTRT, although the radiation pneumonitis incidences were similar.

Comparing HT + CTRT vs. RT alone, a significant survival advantage was also observed with HT at after 1, 2, 3 and 5 years (OR 3.20, 2.09, 2.43 and 3.47, respectively; *p* < 0.05). Lower recurrence (OR 0.39; *p* = 0.0001) and distant metastasis rates (OR 0.46; *p* = 0.003) were recorded, in addition to a higher CR (OR 2.12; *p* = 0.003) and TER (OR 4.8; *p* = 0.002). There was, however, a trend of higher toxicities with HT + CTRT.

### 3.5. Hepatocellular Carcinoma (HCC)

A phase II RCT of 80 patients with primary advanced unresectable HCC was performed [52]. Patients were randomised between the groups of radiofrequency HT + RT vs. RT alone. A capacitive RF system was used. The normalisation of liver enzymes and albumin levels improved more with HT (*p* < 0.05). The therapeutic efficiency (CR, PR or SD) at 3 months was better following HT (60.0% vs. 47.5%; *p* < 0.001). The 1-year recurrence (27.5% vs. 40.0%; *p* < 0.001) and mortality rates (12.5% vs. 20.0%; *p* < 0.001) were also significantly reduced in the HT group.

### 3.6. Pancreatic Cancer

A systemic review compared the addition of HT to RT and/or CT. A total of 14 studies (none were RCTs), consisting of 395 patients with LA or metastatic pancreatic cancer, were analysed [53]. A longer median OS (11.7 vs. 5.6 months) and better ORR (43.9% vs. 35.3%) was reported with HT. Most of the reported toxicities were mild, but there was one case of severe subcutaneous fatty burns. In the review, a prospective open-label comparative cohort was included. In total, 68 patients with LA pancreatic cancers were treated with CTRT+/−HT [54]. The median OS was better with HT (15 vs. 11 months, *p* = 0.025) without increasing toxicities.

### 3.7. Rectal Cancer

A total of 137 rectal cancer patients undergoing NA CTRT were randomised to RF HT (BSD 2000s) [55]. No statistical difference in the global ‘Gastrointestinal Quality of Life Index’ questionnaire at four time points was detected. Response or survival data were not reported, and a trend of increasing toxicity and post-op complications occurred in the HT group. 

A Cochrane Review [56] of pre-operative RT+/−HT in patients with LA rectal cancer used 6 RCTs that comprised 520 patients [28,57,58,59,60]. The 2-year OS was better with HT (HR 2.06; *p* = 0.001), but this difference disappeared after a longer period (3-, 4- and 5-year OS). The CR rates were higher with HT (RR 2.81; *p* = 0.01). Acute toxicity was not different between the treatment arms. Late toxicity data were not reported.

More recently, a matched cohort of 120 LA rectal cancer patients receiving NA CTRT+/−mEHT was reported [61]. In the mEHT (EHY2000) arm, the median RT dose was lower. Larger tumours (>65 cm^3^) showed improved regression (31.6% vs. 0%; *p* = 0.024) and gastrointestinal toxicities were less (64.5% vs. 87.9%; *p* = 0 .01). No difference in the 2-year DFS, OS, LRRFS or DMFS was noted.

### 3.8. Anal Cancer

Ott [62] reported the outcomes of 112 consecutive patients with UICC stage I–IV anal cancer who received CTRT. A total of 50 patients received additional radiative HT (BSD 2000-3D). At the 5-year follow-up point, the OS (95.8% vs. 74.5%; *p* = 0.045), DFS (89.1% vs. 70.4%; *p* = 0.027), LRFS (97.7% vs. 78.7%; *p* = 0.006), and colostomy-free survival rates (87.7% vs. 69.0%; *p* = 0.016) were better with HT. Disease-specific, regional failure-free, and distant metastasis-free survival rates were not different. The adjusted HRs for death (0.25; *p* = 0.036) and local recurrence (0.14; *p* = 0.06) improved with HT. With the exception of haematotoxicity, which was higher with HT (66% vs. 43%; *p* = 0.032), the reported early grade 3–4 toxicities were comparable between treatment arms. The incidences of late side effects were similar, except for a higher telangiectasia rate in HT (38% vs. 16.1%; *p* = 0.009).

The esophageal, HCC, pancreatic, and anorectal cancer articles reviewed above are summarized in Table 3.

### 3.9. Head and Neck Cancers (HNCs) and Nasopharyngeal Carcinomas (NPC) 

Kang [63] reported the outcomes of a phase II RCT using CTRT + HT in the treatment of 154 N2/3 NPC patients. The patients were randomised to microwave HT (Pingliang 778WR-L-4) to the metastatic LN. At 3 months post-treatment, cervical LN CR was better (81.6% vs. 62.8%; *p* = 0.014) with HT. The 5-year LC (96.1% vs. 76.9%; *p* = 0.001), DFS (51.3% vs. 20.5; *p* = 0.001) and OS (68.4% vs. 50.0%, *p* = 0.001) rates were improved with HT. Dermatitis incidence was not significantly higher and no severe complications were observed in any of the patients during the 5-year follow-up. In the patients receiving HT, the 3-month and 5-year LN regressions rates were better if higher temperatures (T90 ≥ 43 °C) or 4–10 sessions were given.

Another phase II RCT compared the outcomes of 83 NPC patients that had definitive CTRT+/−HT [64]. Capacitive RF HT was given using HG-2000/NRL-002 applicators. The median DFS was better with HT (61 vs. 38 months; *p* = 0.048). In addition, 3-year OS was also improved (73.0% vs. 53.5%; *p* = 0.041). Post-treatment NPC-specific QoL scores were also better preserved with HT.

A meta-analysis evaluated the outcomes of HT + RT vs. RT alone in HNCs [65]. A total of 451 cases from 6 studies [8,66,67,68,69,70] were included (five RCTs; one NPC-only trial). No concurrent CT or surgery was used, and RT dose was variable. Overall CR was higher with the addition of HT (39.6% vs. 62.5%; OR 2.92; *p* = 0.001). Acute and late grade 3/4 toxicities were similar in both the groups. Five trials reported long-term survival outcomes using different end points. Patients fared better with HT + RT. The longest survival figures, as reported by Valdagni [68], showed improved 5-year freedom from local relapse (68.6% vs. 24.2%; *p* = 0.015) and OS (53.3% vs. 0%; *p* = 0.02) with HT.

A multicentre phase II Chinese RCT compared the induction of CT + HT vs. CT alone in 120 LA resectable oral squamous cell carcinoma (OSCC) patients [71]. An ultrasonic HT system was used. Treatment was followed by radical surgery and post-operative RT. The clinical response rate was better with HT (65.45% vs. 40.0%; *p* = 0.0088). DFS improved (HR 0.5671; *p* = 0.0335), but not OS (HR 0.6022; *p* = 0.0551). No unexpected toxicity or increase in perioperative morbidity was noted. A 3.33% grade 1/2 skin toxicity rate was associated with HT. OS and DFS were associated with better clinical response in the subgroup analysis.

The HNCs and NPCs articles discussed are summarized in Table 4.

### 3.10. Soft Tissue Sarcoma (STS) 

A total of 341 patients with localised high-risk STS were randomised to NACT+/−RHT (BSD-2000 system) in the EORTC 62961-ESHO 95 multicentre phase III RCT [72,73]. Patients were stratified according to presentation, centre and site. In patients with extremity sarcomas, higher treatment responses (28.8% vs. 12.7%; *p* = 0.002) and R0 resection rates were observed with combined treatment. Similarly, better response rates were observed in retroperitoneal and abdominal STS groups (34.7% vs. 15.6%; *p* = 0.034) [74]. Patients who received HT had better LPFS (2 year: 76% vs. 61%; HR 0.58; *p* = 0.003) and DFS (HR 0.70; *p* = 0.0011). In per-protocol analysis, the HT group had better OS (HR 0.66, *p* = 0.038) [73]. After longer follow-ups (>11 years), further separation of the survival curves was noted. HT improved median LPFS (67.3 vs. 29.2 months; RH = 0.65, *p* = 0.002), median DFS (7.4 vs. 33.3 months; HR = 0.71, *p* = 0.01) and median (15.4 vs. 6.2 years, HR = 0.73; *p* = 0.04) 5-year (62.7% vs. 51.3%) and 10-year OS (52.6% vs. 42.7%). The survival benefit of RHT was noted across all subgroups. Five deaths (3.1%) were attributable to treatment in the combined group vs. two deaths (1.2%) in the NACT-alone group [72]. Toxicities, e.g., leukopenia (grade 3/4), were more frequent with HT (77.6% vs. 63.5%; *p* = 0.005). HT-related grade 3–4 AEs were as follows: 4.3% pain, 4.9% bolus pressure, and 0.6% skin burns [73]. 

Out of 94 patients with macroscopically resected retroperitoneal or abdominal STS, early progression occurred in 10 patients (22.2%) treated with NACT only vs. none with RHT (*p* < 0.001). In addition, 5-year LPFS (56% vs. 45%; *p* = 0.044) and DFS (34% vs. 27%; *p* = 0.040) improved with RHT. OS, perioperative morbidity, and mortality were not different between arms [74].

Immune infiltrates in the biopsies at baseline and after induction treatment were analysed in 109 patients. Post-treatment high tumour-infiltrating lymphocytes (TILs) correlated with better LPFS. A strong association between high TILs or CD8 T cell infiltration and tumour response was noted for patients receiving RHT (*p* = 0.02), but not for the control. It was concluded that HT appeared to prime the tumour microenvironment, probably enabling enhanced anti-tumour immune activity in high-risk STS [75].

### 3.11. Bladder

A Dutch multicentre prospective RCT was performed in 101 muscle-invasive bladder cancer (MIBC) patients, who were randomised to RT vs. HT + RT [28]. HT was given using various radiative RF systems. Improved CR was noted with HT (73% vs. 51%; *p* = 0.01). However, at 3 years, the difference in LC and OS was non-significant.

### 3.12. Glioma 

In a prospective case–control study, 38 glioblastoma patients underwent CTRT or CTRT + HT [76]. HT was given via a capacitive system (Celsius 42+). Pre- (*_V1_*) and post- (*_V2_*) treatment MRI comparisons showed improvements in tumour reduction (ratio (^*V2*^/_*V1*_) 1.12 vs. 0.66 at 6 months) in favour of HT. The OS at 15 months and performance score change was not significantly different between the groups. HT was well tolerated without any significant AEs.

The STS, Bladder cancer and Gliomas articles discussed above are summarized in Table 5.

### 3.13. Palliation

In total, 108 patients with incurable superficial lesions <3 cm from the surface were randomised to RT+/−HT [77]. HT was given using microwave spiral strip applicators. CR improved with HT (66.1% vs. 42.3%; OR 2.7, *p* = 0.02). Previously irradiated patients had the greatest incremental gain in CR (68.2% vs. 23.5%). HT was generally well tolerated, but a higher portion of grade 1–3 skin burn toxicities (46% vs. 5.7%), with one patient having a third-degree skin burn, was observed. No OS benefit was noted.

A Chinese trial compared local mEHT (EHY2000) in combination with traditional Chinese medicines (TCM) vs. the control of intraperitoneal chemoinfusion (IPCI) for the palliation of peritoneal carcinomatosis with malignant ascites (PCMA) [78]. A total of 260 patients were randomized between the 2 arms. In the experimental arm, superior ORR (77.69% vs. 63.85%; *p* < 0.05), QoL scores (48.23% vs. 32.3%; *p* < 0.05) and lower adverse reactions rates (2.3% vs. 12.3%; *p* < 0.05) were observed. All the AEs were grade 1. No survival data were reported.

Furthermore, 103 patients with multiple liver metastases from breast cancer were assigned to CT+/−RHT (MagTherm) [79]. Higher therapeutic efficacy (PR + SD) (75.9% vs. 42%, *p* < 0.01) and QoL scores were noted with RHT. The median time to progression was prolonged with RHT (8.51 vs. 4.32 months; *p* < 0.05) and no serious AEs were reported.

A total of 57 patients with painful bone metastases were randomised to RT (30Gy/10#)+/−HT (Thermotron RF-8) in a phase III RCT [80]. Improved complete pain responses (37.9% vs. 7.1%; *p* = 0.006) and pain control durability (28 days vs. not reached (NR); *p* < 0.001) were observed with HT. QoL improved in the first month, but not the third month. No change in skin or grade ≥3 toxicities were noted. However, 48.3% reported mild heating pain and 20.6% had elevated body temperatures that were resolved shortly after. Obese patients were more likely to experience subcutaneous fat induration. The trial was stopped following interim analysis due to a significant clinical effectiveness and slow recruitment.

The palliative articles discussed above are summarized in Table 6.

## 4. Discussion

A total of 31 articles that used RHT across various cancer types are reviewed here. These include 9 systematic reviews and meta-analyses, and 22 prospective trials (16 randomised trials) published between January 2000 and November 2022. Trials before 2000, although informative, were not reviewed, as their treatment practices may not be current and applicable. Retrospective studies and case series, although important, were also not included in our review, due to an inherent risk of confounding biases.

In these studies, HT was deployed using a variety of technologies and settings. Nevertheless, a general trend of improvements in the therapeutic effects, such as the tumour response, local control rates and QoL outcomes (notably pain), can be observed when RHT is added to standard treatments. Reassuringly, the overall severe toxicity rates are not increased. However, low-grade and temporary skin or pain toxicities were higher in several studies. 

Importantly, several studies also report significant improvements in OS when RHT was employed. For example, a meta-analysis by Yea reports improved 5-year OS with HT in LACC patients undergoing radical CTRT [34]. In a multi-national phase III RCT with patients undergoing NA CT for STS, the median OS more than doubled with minimal toxicities [72], resulting in HT being included in both the NCCN and ESMO guidelines. Other presented studies also point to the potential of using RHT alone, as an immune stimulator and even allowing for treatment de-escalations.

Whilst these findings are encouraging, one should be circumspect when interpreting the results. Several of the meta-analyses had combined studies that spanned a wide duration (including older studies before 2000), using different study types (observational and RCTs), and included trial data that are not publicly available. This could result in significant heterogeneity of the patient cohorts and interventions, which may compromise the validity and applicability of the results. There is also a risk of publication bias. Despite this, the data do present an estimate of the true effect.

The limitations of our narrative review were that other methods, such as whole-body HT and interstitial/intracavitary HT, were not reviewed. Retrospective and single-armed cohort studies were also not included. The methodology of the trials reported here was also not formally assessed for quality and the results were not synthesised, which precludes us from drawing any firm conclusions. However, the purpose of our review is to identify and present higher-level reports that would provide oncologists with a broad overview of RHT in the adjunctive management of cancer.

In conclusion, the efficaciousness of RHT as an adjunct to modern cancer treatments appears promising. It is encouraging to note that there is an increasing amount of research on this subject, with most of the presented reports herein published within the last decade. Although limited, there is some high-quality clinical evidence that RHT offers benefits in certain scenarios, and more RCTs are needed.

## Figures and Tables

**Figure 1 cancers-15-00346-f001:**
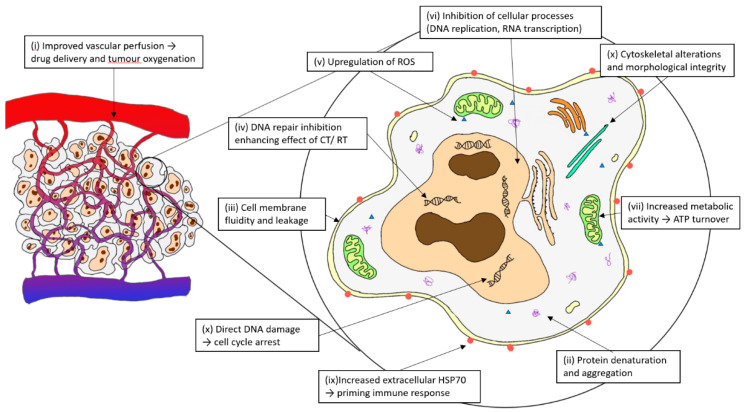
Biological mechanisms of hyperthermia.

**Table 1 cancers-15-00346-t001:** HT in patients with cervical cancers. Summary of articles reviewed.

Author	Article Type	Investigation	Total Participants	Survival Outcome
Lutgens et al., 2010 [22]	Cochrane Systemic Review	HT + RT vs. RT alone in LACC	N = 487 (6 RCTs)	Improved CR, local recurrence rate, and better OS (HR 0.67; *p* = 0.05).
Datta et al., 2016 [29]	NMA	HT + RT+/−CT vs. RT+/−CT in LACC	N = 1160 (16 RCTs)	HT + RT was superior to RT alone in CR and LRC. Non-significant OS benefit. HT + CTRT resulted in best SUCRA score.
Datta et al., 2019 [30]	NMA	Compared across 13 interventional options in LACC	N = 9894 (59 RCTs)	Top 3 interventions by SUCRA: LRC: HT + RT, CTRT + adjCT and HT + CTRT. OS: CTRT (3-weekly CDDP), HT + CTRT and CTRT (non-CDDP). Cumulative: HT + RT, HT + CTRT and CTRT.
Minnaar et al., 2019 [36,37]	Phase III RCT	mEHT + CTRT vs. CTRT in LACC	N = 210	Better 6-month LDC; 2- and 3-year DFS. No OS benefit (except for FIGO III). Better QoL data with mEHT.
Yea et al., 2021 [34]	Meta-analysis	HT + CTRT vs. CTRT in LACC	N = 536 (2 RCTs)	Improved OS (HR 0.67; *p* = 0.03). No LRFS benefit.
Lee et al., 2017 [40]	Prospective comparative trial	mEHT + CT vs. CT in recurrent cervical cancer	N = 38	ORR improved. No OS benefit.

**Table 2 cancers-15-00346-t002:** HT in patients with breast and lung cancers. Summary of articles reviewed.

Author	Article Type	Investigation	Total Participants	Survival Outcome
Breast Cancer				
Datta et al., 2016 [41]	Meta-analysis	RT vs. HT + RT in local recurrent breast cancer	N = 627 (5 RCTs, 3 cohort trials)	CR improved with HT. No survival data reported.
Loboda et al., 2020 [43]	Phase II RCT	NACT + HT vs. NACT in stage IIB–IIIA breast cancer	N = 200	Better tumour and axillary LN size reduction. Increased objective response. Higher 10-year OS rates (*p* = 0.009).
Lung Cancer				
Mitsumori et al., 2007 [44]	Phase II RCT	HT + RT vs. RT alone in LA NSCLC	N = 80	No difference in response rates or OS. Improved PFS (1-year 29.0% vs. 67.5%).
Shen et al., 2011 [45]	Phase II RCT	CT + HT vs. CT alone in advanced NSCLC	N = 80	No change in response rates. Better QoL improvements (especially pain response) with HT.
Ou et al., 2020 [46]	Phase II RCT	IV VitC + meHT vs. BSC in advanced NSCLC	N = 97	Improved disease control rate. Prolonged PFS. Better OS (9.4 m vs. 5.6 m; *p* < 0.05). Better QoL outcomes.
Lee et al., 2013 [47]	Prospective comparative trial	CT + mEHT vs. CT alone in SCLC	N = 31	Improved survival (*p* < 0.02).

**Table 3 cancers-15-00346-t003:** HT in patients with gastrointestinal and hepato-pancreatic cancers. Summary of articles reviewed.

Author	Article Type	Investigation	Total Participants	Survival Outcome
Oesophageal Cancer				
Hu et al., 2017 [48]	Meta-analysis	HT + CTRT vs. CTRT or RT alone	N = 1519 (19 RCTs)	HT + CTRT vs. CTRT: better CR and TER. No difference in recurrence and distal metastases rates. Improved 1-, 3-, 5- and 7-year OS. HT + CTRT vs. RT alone: better CR and TER. Lower recurrence and distal metastases rates. Improved 1-, 2-, 3- and 5-year OS.
HCC				
Dong et al., 2016 [52]	Phase II RCT	HT + RT vs. RT alone in advanced HCC	N = 80	Improved liver enzyme and TER. Reduced recurrence rates. Reduced 1-year mortality (12.5% vs. 20.0%; *p* < 0.001).
Pancreatic Cancer				
Van de Horst et al., 2017 [53]	Systematic review	Addition of HT to RT and/or CT	N = 395 (14 cohort trials)	Improved median OS and ORR, but not statistically analysed.
Maluta et al., 2011 [54]	Prospective comparative trial	HT + CTRT vs. CTRT in LA pancreas cancer	N = 68	Improved median OS (*p* = 0.025).
Rectal Cancer				
Schulze et al., 2006 [55]	Phase II RCT	HT + CTRT vs. CTRT in NA rectal cancer	N = 137	No difference in QoL. No survival/response data.
Haas-Kock et al., 2009 [56]	Cochrane Systematic Review	HT + RT vs. RT alone in NA rectal cancer	N = 520 (6 RCTs)	CR higher (RR 2.81; *p* = 0.01). 2-year OS improved (HR 2.06; *p* = 0.001), but not for 3-,4- or 5-year OS.
Kim et al., 2021 [61]	Prospective comparative trial	mEHT + CTRT vs. CTRT in NA rectal cancer	N = 120	More regression in large tumours. No difference in DFS, OS, recurrent or distal metastases rates.
Anal Cancer				
Ott et al., 2018 [62]	Prospective comparative trial	HT + CTRT vs. CTRT alone	N = 112	No difference in regional failure and distal metastases. Improved 5-year DFS, LRFS and OS (95.8% vs. 74.5%; *p* = 0.045).

**Table 4 cancers-15-00346-t004:** HT in patients with head and neck cancers, including NPC. Summary of articles reviewed.

Author	Article Type	Investigation	Total Participants	Survival Outcome
Kang et al., 2013 [63]	Phase II RCT	HT + CTRT vs. CTRT in N2-3 NPC	N = 154	Improved 3-month CR. 5-year LCR and DFS better. 3- and 5-year OS improved.
Zhao et al., 2014 [64]	Phase II RCT	HT + CTRT vs. CTRT in NPC	N = 83	DFS improved. 3-year OS better (73.0% vs. 53.5%; *p* = 0.041). Better QoL preservation.
Datta et al., 2016 [65]	Meta-analysis	HT + RT vs. RT in HNCs	N = 451 (5RCTs; 1 non-RCT)	Improved overall CR (OR = 2.92; *p* = 0.001). Survival not analysed.
Ren et al., 2021 [71]	Phase II RCT	Induction CT + HT vs. CT alone in OSCC	N = 120	Improved clinical response rates. Improved DFS (HR 0.57; *p* = 0.034). No significant OS advantage.

**Table 5 cancers-15-00346-t005:** HT in patients with soft tissue sarcoma, bladder cancer and glioma. Summary of articles reviewed.

Author	Article Type	Investigation	Total Participants	Survival Outcome
Soft Tissue Sarcoma				
Issels et al., 2018 [72]	Phase III RCT	HT + NACT vs. NACT alone in localised high-risk STS	N = 341	Improved response. Improved LPFS and DFS. OS improved (HR = 0.73; *p* = 0.04).
Bladder Cancer				
Van der Zee et al., 2000 [28]	Phase III RCT	HT + RT vs. RT alone in advanced pelvic tumours	N = 101 (MIBC)	Improved CR. No difference in OS and LC rates.
Glioma				
Mahdavi et al., 2020 [76]	Prospective comparative trial	CTRT vs. CTRT + HT in glioblastoma	N = 38	Improved response. No difference in OS or performance score change.

**Table 6 cancers-15-00346-t006:** HT for patients with palliation. Summary of articles reviewed.

Author	Article Type	Investigation	Total Participants	Survival Outcome
Jones et al., 2005 [77]	Phase II RCT	RT + HT vs. RT alone in superficial skin tumours	N = 108	Improved CR. No OS benefit.
Pang et al., 2017 [78]	Phase II RCT	mEHT + TCM vs. IPCI in PCMA	N = 260	Higher ORR. Improved QoL.
Kilmanov et al., 2018 [79]	Phase II RCT	HT + CT vs. CT alone in breast cancer and liver metastases	N = 103	Higher PR and SD. Improved QoL. Longer median time to progression.
Chi et al., 2018 [80]	Phase III RCT	RT + HT vs. RT alone in painful bone metastases	N = 108	Improved pain response. Longer time to pain progression. Improved 1-month QoL.

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
