# Peer review of "A Review of the Current Clinical Evidence for Loco-Regional Moderate Hyperthermia in the Adjunct Management of Cancers"

_cancers, 2023, doi:10.3390/cancers15020346_

Round 1

Reviewer 1 Report

This manuscript is a narrative review that summarizes a total of 30 recently published studies to present high-quality prospective trial data on the clinical effects of regional hyperthermia therapy (RHT) as an adjunct to standard care for various cancers. Studies included in this narrative review are mainly English-language full-text clinical trials, meta-analyses, and systematic reviews. Authors highlight various benefits related to control, survival, and quality of life to show the potential of RHT as adjunctive therapy for cancer.

This manuscript may positively be described as interesting and deserving for publication. However, I would appreciate it if the following comments could be taken into account:

1. As the manuscript contains too many abbreviations, it would be nice if you could tabulate them before the results section.

2. In the introduction it would be more informative to elaborate more on the biological mechanism of action of RHT and to highlight its advantages over, for example, whole-body hyperthermia in the management of solid tumors in their non-metastatic stages.

3. If and where applicable, indicate what type of technologies were used for RHT application in the reported studies.

4. Regarding Table 1, it would be more understandable to split this table into multiple tables and place them under their respective sections in the results section rather than combining everything into one table.

5. The references are all numbered twice. Please correct this.

Thank you and best of luck!

Author Response

Dear Reviewer,

Thank you for the in-depth review and comments. Several amendments have been made as suggested and I believe these would significantly improve on the readability and appreciation of this article. 

With regard to the following points, the following changes have been made.

Point 1: As the manuscript contains too many abbreviations, it would be nice if you could tabulate them before the results section.

Response 1: An abbreviation section has been added. But the location of this was at the end of the article per journal format.

Point 2: In the introduction, it would be more informative to elaborate more on the biological mechanism of action of RHT and to highlight its advantages over, for example, whole-body hyperthermia in the management of solid tumors in their non-metastatic stages.

Response 2: This paragraph has been edited quite extensively and a figure has been added to illustrate the mechanisms of action of hyperthermia. Other examples of hyperthermia methods have been given but were not focused on, as we wanted to review only loco-regional hyperthermia, however references have been made for readers to refer to the types of hyperthermia available as well as the mechanisms of action.

Point 3: If and where applicable, indicate what type of technologies were used for RHT application in the reported studies.

Response 3: Where available, the technology including the machine used has been added to the relevant study paragraphs.

Point 4: Regarding Table 1, it would be more understandable to split this table into multiple tables and place them under their respective sections in the results section rather than combining everything into one table.

Response 4: Done. Thank you for this suggestion. We have split the table up into 5 tables for easier reading.

Point 5: The references are all numbered twice. Please correct this.

Response 5: Thank you for pointing this out. This must have happened with the formatting after submission. It has been rectified.

Thank you for the very insightful recommendations and comments. I hope you would find the changes made acceptable and improve the overall readability and relevance of this article.

Best Regards.

Reviewer 2 Report

The authors present a review of clinical evidence for hyperthermia achieved in the past two decades.

The papers included seem fairly complete, I may suggest a few more for missing tumor sites, and a few more for the interpretation of the data.

Line 43/44: The introduction is carefully listing what is known of the biological rationale of hyperthermia, and the failure of some early American trials with absent quality assurance. It is a bit odd to mention the failures and not the positive trials in the sentence ‘Unfortunately, robust clinical data remains elusive and marred by early negative trials’, which I would change into ‘Unfortunately, though positive results have been reported robust clinical data remains elusive and marred by early negative trials’, where you can refer to existing ref 7 (Peeken) and maybe also another review paper for the positive trials as reference:

Datta NR, et al. Local hyperthermia combined with radiotherapy and-/or chemotherapy: recent advances and promises for the future. Cancer Treat Rev. 2015 Nov;41(9):742-53. doi: 10.1016/j.ctrv.2015.05.009. PMID: 26051911

You should remove references 11 and 12: 11 Roussakov because this author has no scientific track record and published only in the company journal of oncotherm, vendor of mEHT hyperthermia, and 12 because the link does not work, and a post would be a strange reference anyway.

Instead you should refer to an excellent review on reasons for frequent failure of hyperthermia:

Liebl CM, et al. Systematic review about complementary medical hyperthermia in oncology. Clin Exp Med. 2022 Nov;22(4):519-565. doi: 10.1007/s10238-022-00846-9. PMID: 35767077

Liebl et al identify the lack of temperature monitoring in many trials as an important reason for failure .

Line 57-58: you state ‘Heat distribution is calculated and can be temperature monitored real-time by thermometric probes.’ QA guidelines are very clear that real-time temperature monitoring is mandatory in RHT, so this should be rephrased to: ‘The heat distribution is calculated and target temperatures are monitored real-time by minimally invasive thermometric probes.’

Line 58-60: thermal dosimetry, please add references on definition for CEM43:

Sapareto SA, Dewey WC. Thermal dose determination in cancer therapy. Int J Radiat Oncol Biol Phys. 1984 Jun;10(6):787-800. doi: 10.1016/0360-3016(84)90379-1. PMID: 6547421

Please add after the sentence on thermal dosimetry (line 58-60) a sentence on the need to achieve an adequate therapeutic dose, for instance: ‘Achieving an adequate temperature rise is important of the clinically found dose-effect relationship’. References should include your ref 34 (Ohguri et al 2018), and:

Kroesen M, et al. Confirmation of thermal dose as a predictor of local control in cervical carcinoma patients treated with state-of-the-art radiation therapy and hyperthermia. Radiother Oncol. 2019 Nov;140:150-158. doi: 10.1016/j.radonc.2019.06.021. PMID: 31302345

Bakker A, et al. Temperature and thermal dose during radiotherapy and hyperthermia for recurrent breast cancer are related to clinical outcome and thermal toxicity: a systematic review. Int J Hyperthermia. 2019;36(1):1024-1039. doi: 10.1080/02656736.2019.1665718. PMID: 31621437

Nakahara S, et al. Intensity-Modulated Radiotherapy with Regional Hyperthermia for High-Risk Localized Prostate Carcinoma. Cancers (Basel). 2022 Jan 13;14(2):400. doi: 10.3390/cancers14020400. PMID: 35053562 [this study is also interesting as it covers results for prostate, presently absent in your review!]

Line 60-62: no one has ever been able to show that the claimed mechanism of mEHT actually works, so you should change the sentence to read: ‘A variation of RHT, modulated electro hyperthermia (mEHT), uses nonhomogeneous heating which is claimed to cause destabilization of malignant cell membranes [18].’

The table with 30 studies is impressive, but I think I am missing some tumor sites, like prostate:

Yahara K, et al. Definitive radiotherapy plus regional hyperthermia for high-risk and very high-risk prostate carcinoma: Thermal parameters correlated with biochemical relapse-free survival. Int J Hyperthermia. 2015:31:6, 600-608, DOI: 10.3109/02656736.2015.1062214

I already mentioned:

Nakahara S, et al. Intensity-Modulated Radiotherapy with Regional Hyperthermia for High-Risk Localized Prostate Carcinoma. Cancers (Basel). 2022 Jan 13;14(2):400. doi: 10.3390/cancers14020400. PMID: 35053562

These issues need to be resolved by the authors before this manuscript can be deemed acceptable for publication

Author Response

Dear Reviewer,

Thank you for the in-depth review and comments. Several amendments have been made as suggested and I believe these would significantly improve on the readability and appreciation of this article. 

With regard to the following points, the following changes have been made.

Point 1: Line 43/44: The introduction is carefully listing what is known of the biological rationale of hyperthermia, and the failure of some early American trials with absent quality assurance. It is a bit odd to mention the failures and not the positive trials in the sentence ‘Unfortunately, robust clinical data remains elusive and marred by early negative trials’, which I would change into ‘Unfortunately, though positive results have been reported robust clinical data remains elusive and marred by early negative trials’, where you can refer to existing ref 7 (Peeken) and maybe also another review paper for the positive trials as reference:

Datta NR, et al. Local hyperthermia combined with radiotherapy and-/or chemotherapy: recent advances and promises for the future. Cancer Treat Rev. 2015 Nov;41(9):742-53. doi: 10.1016/j.ctrv.2015.05.009. PMID: 26051911

You should remove references 11 and 12: 11 Roussakov because this author has no scientific track record and published only in the company journal of oncotherm, vendor of mEHT hyperthermia, and 12 because the link does not work, and a post would be a strange reference anyway.

Instead you should refer to an excellent review on reasons for frequent failure of hyperthermia:

Liebl CM, et al. Systematic review about complementary medical hyperthermia in oncology. Clin Exp Med. 2022 Nov;22(4):519-565. doi: 10.1007/s10238-022-00846-9. PMID: 35767077

Liebl et al identify the lack of temperature monitoring in many trials as an important reason for failure .

Line 57-58: you state ‘Heat distribution is calculated and can be temperature monitored real-time by thermometric probes.’ QA guidelines are very clear that real-time temperature monitoring is mandatory in RHT, so this should be rephrased to: ‘The heat distribution is calculated and target temperatures are monitored real-time by minimally invasive thermometric probes.’

Response 1: Thank you for these recommendations. We fully agree with them and have amended the article and references as suggested.

Point 2: Line 58-60: thermal dosimetry, please add references on definition for CEM43:

Sapareto SA, Dewey WC. Thermal dose determination in cancer therapy. Int J Radiat Oncol Biol Phys. 1984 Jun;10(6):787-800. doi: 10.1016/0360-3016(84)90379-1. PMID: 6547421

Please add after the sentence on thermal dosimetry (line 58-60) a sentence on the need to achieve an adequate therapeutic dose, for instance: ‘Achieving an adequate temperature rise is important of the clinically found dose-effect relationship’. References should include your ref 34 (Ohguri et al 2018), and:

Kroesen M, et al. Confirmation of thermal dose as a predictor of local control in cervical carcinoma patients treated with state-of-the-art radiation therapy and hyperthermia. Radiother Oncol. 2019 Nov;140:150-158. doi: 10.1016/j.radonc.2019.06.021. PMID: 31302345

Bakker A, et al. Temperature and thermal dose during radiotherapy and hyperthermia for recurrent breast cancer are related to clinical outcome and thermal toxicity: a systematic review. Int J Hyperthermia. 2019;36(1):1024-1039. doi: 10.1080/02656736.2019.1665718. PMID: 31621437

Nakahara S, et al. Intensity-Modulated Radiotherapy with Regional Hyperthermia for High-Risk Localized Prostate Carcinoma. Cancers (Basel). 2022 Jan 13;14(2):400. doi: 10.3390/cancers14020400. PMID: 35053562 [this study is also interesting as it covers results for prostate, presently absent in your review!]

Response 2: Thank you for the comments and providing the suggested references. This paragraph has been edited to highlight the importance of good temperature targets and monitoring.

“Heat distribution is calculated and target temperatures are monitored real-time by minimally invasive thermometric probes. An adequate temperature rise is important to achieve a good clinical outcome and a dose-effect relationship has been reported in many studies[17–19]. Thermal dosimetry is thus quantified by temperature and duration and expressed in cumulative equivalent minutes at a temperature of 430C (CEM43)[20] and Tx, which represents the temperature exceeded by X% of the intra-tumor points.”

Point 3: Line 60-62: no one has ever been able to show that the claimed mechanism of mEHT actually works, so you should change the sentence to read: ‘A variation of RHT, modulated electro hyperthermia (mEHT), uses nonhomogeneous heating which is claimed to cause destabilization of malignant cell membranes [18].’

Response 3: We understand the need to change and have made the needed amendments.

“A variation of RHT, modulated electro hyperthermia (mEHT), uses nonhomogeneous heating which is claimed to cause destabilization of malignant cell membranes [21].”

Point 4: The table with 30 studies is impressive, but I think I am missing some tumor sites, like prostate:

Yahara K, et al. Definitive radiotherapy plus regional hyperthermia for high-risk and very high-risk prostate carcinoma: Thermal parameters correlated with biochemical relapse-free survival. Int J Hyperthermia. 2015:31:6, 600-608, DOI: 10.3109/02656736.2015.1062214

I already mentioned:

Nakahara S, et al. Intensity-Modulated Radiotherapy with Regional Hyperthermia for High-Risk Localized Prostate Carcinoma. Cancers (Basel). 2022 Jan 13;14(2):400. doi: 10.3390/cancers14020400. PMID: 35053562

Response 4: Thank you for these suggestions. These articles mentioned are indeed useful and important. However, the studies mentioned were both retrospectively conducted and we focused only on prospective trials in our article (line 14/15 of highlights, line 24 of abstract, line 75/76 of Methods).

In the review article, you had previously brought to our attention by Liebl et al. there was indeed a missing article that was not found in our PubMed search.

Mahdavi SR, Khalafi L, Nikoofar AR, Fadavi P, Arbabi Kalateh F, Aryafar T, et al. Thermal enhancement effect on chemo-radiation of glioblastoma multiform. International Journal of Radiation Research. 2020;18:255–62.

However, as the article has been brought to our attention we have added a Glioma subsection for this paper resulting in 31 articles.

In our search, we could not find any prospective-comparative trial evidence for hyperthermia use in Prostate cancer. A recent review by Guevelou et al. also highlighted this need.

Jennifer Le Guevelou, Monica Emilia Chirila, Vérane Achard, Pauline Coralie Guillemin, Orane Lorton, Johannes W. E. Uiterwijk, Giovanna Dipasquale, Rares Salomir & Thomas Zilli (2022) Combined hyperthermia and radiotherapy for prostate cancer: a systematic review, International Journal of Hyperthermia, 39:1, 547-556,

The limitations of this study in excluding retrospective evidence have thus been elaborated in our discussion:

Line 322-324: Retrospective studies and case series, though important, were also not included in our review due to an inherent risk of confounding biases.

Line 344-348: The limitations of our narrative review was that other methods such as whole-body HT, interstitial/ intracavitary HT were not reviewed. Retrospective and single-armed cohort studies were also not included as well. The methodology of the trials reported here were also not formally assessed for quality and the results were not synthesised, which precludes us from drawing any firm conclusions. However, the purpose of our review is to identify and present higher level reports that would provide the oncologist a broad overview of RHT in the adjunctive management of cancer.

Thank you for the very insightful recommendations and comments. I hope you would find the changes made acceptable and improve the overall readability and relevance of this article.

Best Regards.

Round 2

Reviewer 2 Report

The authors have revised their manuscript in an adequate fashion. These revisions are satisfactory. I was wondering whether the authors had seen the RCT by Wang et al for locally advanced cervical cancer:

Wang Y, et al. Outcomes for Hyperthermia Combined with Concurrent Radiochemotherapy for Patients with Cervical Cancer. Int J Radiat Oncol Biol Phys. 2020 Jul 1;107(3):499-511

although probably included in the meta analysis of Yea, your ref 34. A relevant recent review for LACC is:

IJff M, et al. The role of hyperthermia in the treatment of locally advanced cervical cancer: a comprehensive review. Int J Gynecol Cancer. 2022 Mar;32(3):288-296

On reading the newly added sections of the revised manuscript I recommend adding/changing some text and a few more references at key locations:

Line 43: add to ref 2-4 a review that was the first to visualize the hallmarks of hyperthermia in this fashion as in the figure:

Issels R, et al. Hallmarks of hyperthermia in driving the future of clinical hyperthermia as targeted therapy: translation into clinical application. Int J Hyperthermia. 2016;32(1):89-95

Line 65: add to ref 16 another review that pays more attention to capacitive hyperthermia devices:

Kok HP, et al. Heating technology for malignant tumors: a review. Int J Hyperthermia. 2020;37(1):711-741

Line 151: please add ‘the inductive’ in ‘HT was given using the inductive MagTherm device.’

Line 160: please add ‘the capacitive’ in ‘HT was given using the capacitive RF-8 Thermotron device.’

Line 250: add ‘capacitive’ in the line ‘Capacitive RF HT was given using HG-2000/ NRL-002 applicators.’

Line 302: please add ‘radiative’ in ‘HT was given using various radiative RF systems.’

Line 362: replace ‘openly’ with ‘publicly’ in ‘not openly available’

Line 366: rewrite ‘was that other’ into ‘were that other’

Line 368: replace ‘were not included as well’ with ‘were also not included’

These last issues need to be resolved by the authors before this manuscript can be deemed acceptable for publication

Author Response

We thank the reviewer for a very thorough review and quick response.

With regards to the following points/ suggestions, the following changes have been made.

Point 1: I was wondering whether the authors had seen the RCT by Wang et al for locally advanced cervical cancer:

Wang Y, et al. Outcomes for Hyperthermia Combined with Concurrent Radiochemotherapy for Patients with Cervical Cancer. Int J Radiat Oncol Biol Phys. 2020 Jul 1;107(3):499-511

although probably included in the meta analysis of Yea, your ref 34. 

Response 1: The article by Wang et al. was included in the meta-analysis. however you have rightly highlighted it as an important study and thus we have expanded on the paragraph to elaborate the trial results further. However, we did not include it in the summary table as we did not want to repeat the result already represented in the meta-analysis by yea et al. The paragraph has been amended as such.

"More recently, Yea conducted a meta-analysis comparing radical HT+CTRT vs. CTRT alone [36]. 2 RCTs[33, 37], included 536 patients with LACC. Both trials used a RF capacitive heating device (Thermotron RF-8 and NRL-004 device). Harima reported a better CR with HT in his trial (Odds Ratio (OR) 3.993, p=0.047) although OS, disease free survival (DFS) and local relapse free survival (LRFS) improvements were not significant[33]. Wang reported a 5-year OS improvement (81.9% vs 72.3%, p=0.04) though LRFS was not significantly improved[37]. In the combined trial data, 5-year OS (HR 0.67; p=0.03) was better in the group receiving HT although the LRFS  improvement remained not statistically significant (HR 0.74; p=0.16)[36]. Toxicity rates were not different between the arms. Amongst patients who received HT, a higher CEM43T90 (≥1 min) was associated with better LRFS[19]."

Point 2: A relevant recent review for LACC is:

IJff M, et al. The role of hyperthermia in the treatment of locally advanced cervical cancer: a comprehensive review. Int J Gynecol Cancer. 2022 Mar;32(3):288-296

Response 2:

Thank you, HT use has been reviewed most extensively in cervical cancers. this recent review by Ijff is very appropriate and will help explain and guide the use of hyperthermia in cervical cancers to the readers.

We have thus included it as the last paragraph at the end of the cervical cancer subsection, line 135/136 as follows

"We also highlight a recent reviewed by Ijff et al, that provides further explanation and guidance on the use of RHT in LACC[43]."

Point 3: On reading the newly added sections of the revised manuscript I recommend adding/changing some text and a few more references at key locations:

Line 43: add to ref 2-4 a review that was the first to visualize the hallmarks of hyperthermia in this fashion as in the figure:

Issels R, et al. Hallmarks of hyperthermia in driving the future of clinical hyperthermia as targeted therapy: translation into clinical application. Int J Hyperthermia. 2016;32(1):89-95

Line 65: add to ref 16 another review that pays more attention to capacitive hyperthermia devices:

Kok HP, et al. Heating technology for malignant tumors: a review. Int J Hyperthermia. 2020;37(1):711-741

Line 151: please add ‘the inductive’ in ‘HT was given using the inductive MagTherm device.’

Line 160: please add ‘the capacitive’ in ‘HT was given using the capacitive RF-8 Thermotron device.’

Line 250: add ‘capacitive’ in the line ‘Capacitive RF HT was given using HG-2000/ NRL-002 applicators.’

Line 302: please add ‘radiative’ in ‘HT was given using various radiative RF systems.’

Line 362: replace ‘openly’ with ‘publicly’ in ‘not openly available’

Line 366: rewrite ‘was that other’ into ‘were that other’

Line 368: replace ‘were not included as well’ with ‘were also not included’

Response 3:

Thank you for the above. The changes have been made as suggested.

Thank you once again for the review. We are grateful for the suggestions made and believe the revisions will be beneficial to the completeness of the article.

Warmest Regards.

Round 3

Reviewer 2 Report

the authors have correctly processed my comments